# Liver Dysfunction with Severe Cholestasis and Coagulation Disorders in the Course of Hemolytic Disease of the Newborn Requiring Chelation Therapy—A Case Report and Review of the Literature

**DOI:** 10.3390/jcm12247645

**Published:** 2023-12-13

**Authors:** Agnieszka Drozdowska-Szymczak, Julia Proczka, Danuta Chrzanowska-Liszewska, Krzysztof Truszkowski, Natalia Mazanowska, Paweł Krajewski

**Affiliations:** 1Department of Neonatology and Neonatal Intensive Care, Institute of Mother and Child, Kasprzaka 17a, 01-211 Warsaw, Poland; 2Department of Obstetrics and Gynecology, Institute of Mother and Child, Kasprzaka 17a, 01-211 Warsaw, Poland

**Keywords:** neonate, cholestasis, hemolytic disease, chelation therapy, liver failure, coagulation disorders, iron overload, deferoxamine, intrauterine blood transfusion

## Abstract

Severe hemolytic disease of the fetus and newborn (HDFN) requiring intrauterine transfusions (IUTs) may cause iron accumulation, resulting in liver damage, which may lead to cholestasis and coagulation disorders. In this article, we reported a case of a female neonate who underwent chelation therapy with a positive outcome, and we reviewed the English and Polish literature on chelation therapy in HDFN available in PubMed. The patient with maximum ferritin concentration above 33,511.2 ng/mL developed liver dysfunction with coagulation disorders requiring multiple transfusions of fresh frozen plasma (FFP), Octaplex^®^ and cryoprecipitate, and hypoalbuminemia treated with numerous albumin infusions. Furthermore, severe cholestasis was observed with direct bilirubin levels up to 33.14 mg/dL. Additionally, the child developed transient myelosuppression with neutropenia, thrombocytopenia, and low reticulocyte count due to several blood transfusions. The differential diagnosis tests were conducted to rule out any causes of hepatic failure other than hemolytic disease of the newborn. This case proves that adequate treatment of severe HDFN with anemia requiring IUT and hepatic failure can lead to positive outcomes with no long-term consequences.

## 1. Introduction

The severity of hemolytic disease of the fetus and newborn (HDFN) varies significantly among individuals. It is possible that mild anemia and hyperbilirubinemia are the only present symptoms. In other cases, fetal anemia requires intrauterine transfusions (IUTs) and may lead to fetal hydrops and intrauterine fetal demise [1,2]. Multiple IUTs increase the risk of preterm rupture of membranes, preterm labor, chorioamnionitis, and cardiac failure due to volume overload [3].

Severe hemolysis occurring in HDFN and multiple IUTs can lead to the accumulation of iron stored in the liver. In neonates with hemolytic disease, ferritin levels may be above normal [4,5,6]. Excess iron accumulation can lead to hepatic dysfunction, coagulation disorders, and cholestasis. These complications are more common in neonates who have undergone intrauterine blood transfusions due to anemia, most often in the course of Rhesus immunization with the anti-D antibodies present in the maternal serum [7]. To avoid serious complications resulting from the toxic effects of iron overload, chelation therapy may be indicated [8].

## 2. Materials and Methods

A review of the articles published between January 1980 and October 2023 was performed to identify case reports and reviews related to chelation therapy in hemolytic disease of the fetus and newborn. Two authors (A.D.-S., J.P.) independently performed a PubMed search, using the keywords: “iron overload”, “intrauterine transfusion”, “hemolytic disease of the fetus and newborn”, and “chelation therapy”. The reference lists of retrieved articles were also reviewed. Only reviews and articles written in English and Polish were included in further analysis.

After the performed literature search, publications were analyzed by title and abstract to select papers for full-text assessment. Following abstract selection, the remaining full-text articles were screened for eligibility. Collected data included year of publication, gestational age at birth, clinical picture, neonatal management, laboratory results, and outcomes.

## 3. Case Description

A female infant, gravida 3, para 3, was born at the Department of Obstetrics and Gynecology of Warsaw Medical University at 33 1/7 weeks gestation, delivered by cesarean section. In the first pregnancy, at 28 weeks, a positive, low anti-D antibody titer was found in the mother’s serum while pregnant with an RhD-positive fetus. A rapid increase in anti-D antibody levels was observed during the second pregnancy, and fetal death was confirmed at 24 weeks gestation.

The index pregnancy was complicated by red cell alloimmunization with anti-D antibody titer 1:4096 and anti-E antibody titer 1:8. Due to fetal anemia, six IUTs of Rh-negative blood were performed. The fetal blood group was determined as B Rh-negative, a positive indirect antiglobulin test (IAT; indirect Coombs test), with a low expression of D antigen.

Pregnancy was also complicated by diet-controlled gestational diabetes mellitus, anemia, and hypothyroidism in the mother. The mother did not receive antenatal corticosteroid therapy.

Cesarean delivery occurred 1.5 h after IUT due to the threat of fetal asphyxia, with fentanyl and pancuronium administered during the procedure. Bloody amniotic fluid was noted during cesarean section, indicating a hemorrhage from the umbilical vein punctured during the procedure.

The child was born non-vigorous, with an Apgar score of 1/2/3/3 points at 1-3-5-10 min after birth, respectively. Cord blood gas results were as follows: pH 6.89; base excess (BE) −15.5 mmol/L; lactates 11 mmol/L. The neonate weighing 1960 g (53rd percentile using Fenton charts) was floppy, pale, without a respiratory drive, and with severe bradycardia due to hemorrhagic shock. The infant required intubation, ventilation, and external cardiac massage.

After birth, due to severe anemia (Hgb 8.2 g/dL, Hct 24%), leukoreduced and irradiated red blood cells (RBC) were transfused. In total, the child required 7 top-up blood transfusions in the first months of life with no need for exchange transfusions.

After numerous transfusions of blood products intra-utero and after birth, transient bone marrow suppression was observed. Erythropoietin was administered with mediocre results—in the first weeks of the patient’s life, reticulocyte count remained low (minimum 0.12%, absolute reticulocyte count 0.005 × 10^6^/µL), only in the 8th week of life an increase in reticulocyte count was obtained (4.71%, absolute reticulocyte count 0.121 × 10^6^/µL). Transient leukopenia with a minimum absolute neutrophil count of 648 cells/µL was observed. According to guidelines, prophylactic antimicrobial therapy was administered. Inflammatory markers and blood culture remained negative. Filgrastim subcutaneous injections were administered to the patient with good response. The number of neutrophils decreased rapidly after treatment withdrawal, and a long-term improvement was obtained only after multiple doses (five in total).

In the first days of life, severe coagulation disorder was observed, and the child developed transient disseminated intravascular coagulation (DIC) with recurrent thrombocytopenia. Four transfusions of fresh frozen plasma (FFP), three of Octaplex^®^, three of cryoprecipitate, and six platelet infusions (due to low platelet count—the lowest observed of 14 × 10^9^/L) were used in the treatment. In the following days, coagulation parameters were stabilized, and the number of platelets gradually increased.

Phototherapy was used for three days as a consequence of hyperbilirubinemia. Due to the severe HDFN, intravenous immunoglobulin G (IVIG) was administered twice.

Severe cholestasis was observed with direct bilirubin levels up to 33.14 mg/dL. Therapy with ursodeoxycholic acid and fat-soluble vitamins (A, D, E, K) commenced. In the course of cholestasis, the patient passed pale stools. Abdominal ultrasonography and abdominal radiograph revealed peritoneal fluid accumulation, hepatosplenomegaly, and enlargement of the gallbladder. Due to hypoalbuminemia (the lowest observed level of 1.7 g/dL) and associated symptoms, the child was transfused with human albumin solution several times. Serum albumin levels normalized. Follow-up ultrasound examinations of the abdominal cavity showed a regression of abnormalities.

Hepatic failure was diagnosed based on the clinical manifestations and the results of additional tests. The most extreme laboratory results in the first 3 months of life are presented in Table 1. Other possible causes of hepatic failure were excluded as the cause of abnormalities. No bile duct pathology was observed.

It was assumed that the clinical picture was due to intrahepatic cholestasis in the course of hemolytic disease associated with iron accumulation in the liver due to multiple transfusions. At 51 days of age, the patient was transferred to the Neonatology, Neonatal Pathology, and Intensive Care Department at The Children’s Memorial Health Institute in Warsaw for further treatment and testing, where she was provided with chelation therapy.

After chelation therapy with deferoxamine, liver function tests were improving, and the treatment continued for four weeks (five days a week) with good tolerance. Symptomatic treatment with ursodeoxycholic acid and fat-soluble vitamins continued as well.

The respiratory failure was managed with mechanical ventilation for two days, followed by nasal continuous positive airway pressure (nCPAP) for the next two days. A high-flow nasal cannula (HFNC) was required until the 17th day of life. Due to circulatory failure, dopamine was administered for two days. In the first days of life, oliguria associated with low GFR (creatinine 2.45 mg/dL) was observed. Initially, furosemide was administered to promote diuresis with urine output gradually returning to normal by the 6th day of life. Abdominal ultrasonography showed blurred delineation of the kidneys with increased cortical echogenicity. Gradual stabilization of renal function was obtained, and follow-up imaging revealed normalization of the image.

Additionally, neonatal glucose screening revealed hypoglycemia on the first day of life, treated with intravenous infusion. Owing to hypothyroidism, the child received levothyroxine—normalization of thyroid function was observed.

Total parenteral nutrition was used in the first days of life due to the patient’s severe condition and symptoms of hypoxia. A fish oil-based lipid emulsion was used to improve liver function. In the following days, expressed milk feeding was well tolerated, and peripheral parenteral nutrition was used until the 14th day of life. Due to poor weight gain, most likely caused by malabsorption in the course of cholestasis, milk fortifiers (HMF, Fantomalt^®^, Nutricia, Hoofddorp, The Netherlands) and hypercaloric formula were used in addition to breast milk, resulting in weight gain.

Brain MRI performed at eight weeks of age at 40 weeks postmenstrual age (PMA) showed brain structures with no significant abnormalities and age-appropriate myelination.

Currently, the patient is in her second year of life. She receives outpatient care from the Neonatology Clinic in the Institute of Mother and Child in Warsaw and the Liver Diseases Clinic in The Children’s Memorial Health Institute in Warsaw. Body weight, head circumference, and height remain above the 50th percentile (according to Fenton). The patient was fed with Infantrini^®^ milk for one year, and she is following dietary guidelines for healthy children. Treatment with ursodeoxycholic acid was maintained until 1 year of age, currently, the child does not receive medication. Direct bilirubin values, ferritin, AST, and ALT remain within normal range. Ultrasonography shows no abnormality of abdominal structures. The child receives physical therapy as mild hypertonia is observed, with lower limbs more affected. Psychomotor development is adequate to corrected age.

## 4. Discussion

The cause of HDFN is the mother’s antibodies against fetal blood cells. These antibodies can be produced against any of 6 antigens of the Rh system (D, d, C, c, E, e), as well as against antigens of other blood group systems that the child inherited from the father and that are not present in mother’s red cells. Maternal antibodies crossing the placenta bind to red cells and cause hemolysis.

Consequently, heme is broken down into bilirubin, which is transported bound to plasma albumin due to its poor solubility in water. Such a compound is called indirect or unconjugated bilirubin. It can cross the blood–brain barrier with a neurotoxic effect. Indirect bilirubin is converted into direct (conjugated) bilirubin in the liver. That water-soluble compound is secreted into the bile and then excreted to the intestine, where bacterial enzymes deconjugate it into bile pigments.

Since bilirubin is excreted through the placenta, it does not affect the fetus. The hepatic enzyme system of the newborn is immature; therefore, it cannot metabolize excessive bilirubin produced in children affected with HDFN. To prevent kernicterus and damage to the central nervous system, phototherapy is used in severe jaundice. An exchange blood transfusion is performed when intensive phototherapy is ineffective. This procedure, if necessary, should be performed in an intensive care unit due to the risk of complications, including hypocalcemia, hyperkalemia, hypoglycemia, hypernatremia, leukopenia, thrombocytopenia, infections, arrhythmias, necrotizing enterocolitis and even death [9,10]. IVIG is rarely used to treat hyperbilirubinemia in hemolytic disease, as this therapy is still controversial [11,12].

As a result of hemolysis, bilirubin and iron are released (Figure 1). Since the human body does not have effective mechanisms for iron excretion, after the saturation of binding proteins (transferrin, ferritin, hemosiderin), iron is deposited in various tissues and organs—especially the liver, heart, and endocrine system (pancreas, thyroid, parathyroid glands, adrenal glands) [8]. By producing reactive oxygen species, iron has detrimental effects on numerous organs, leading to their dysfunction. Elevated serum ferritin concentrations are common in patients with hemolytic disease. Ferritin level can exceed three times the upper limit of the normal range [4,5,6]. Significant accumulation of iron may cause serious complications, such as cardiac disorders due to cardiomyocyte damage (cardiac arrhythmias, cardiomyopathies, sudden cardiac death) and multiple organ failure leading to death [8].

Hemolysis in HDFN results in bilirubin and iron release. Bilirubin binds to albumin to enable its transport to the liver, where glucuronic acid is added. Meanwhile, free iron accumulates in the tissues, mainly in the liver, heart, and endocrine system, resulting in their dysfunction.

Hepatic dysfunction may also occur, resulting in cholestasis, hypoalbuminemia, and coagulation disorders. Elevated direct bilirubin levels are reported in approximately 7–13% of newborns with HDFN [7,11,13]. In the cited publications, the authors used the old definition of cholestasis: direct bilirubin concentration above 2 mg/dL and not above 1 mg/dL, as is the case today [14]. This affects the number of confirmed cholestasis cases, which would be higher if the new definition was used.

Symptoms of cholestasis include jaundice, rarely green discoloration of the skin, hepatomegaly, and pale stools. Direct bilirubin, the concentration of bile acids, gamma-glutamyl transpeptidase (GGT), aminotransferases, and alkaline phosphatase should be monitored [15].

In the described case, due to hemolytic disease with the presence of anti-RhD and anti-RhE antibodies, hyperbilirubinemia with a maximum concentration of total bilirubin of 43.55 mg/dL was observed, as well as cholestasis with a maximum direct bilirubin level of 33.14 mg/dL. The patient had green-colored skin, hepatomegaly, and passed pale stools.

The differential diagnosis for the causes of cholestasis should include bacterial and viral infection (including TORCH), carbohydrate metabolism disorders (galactosemia, hereditary fructose intolerance), fatty acid oxidation disorders, amino acid metabolism disorders (tyrosinemia), lipid storage diseases (glycogen storage disease type IV, Zellweger syndrome), cystic fibrosis, alpha-1 antitrypsin deficiency, Alagille syndrome, inborn errors of bilirubin metabolism (Crigler–Najjar syndrome, Gilbert’s syndrome, Dubin–Johnson syndrome, and Rotor syndrome), hemochromatosis, adrenal insufficiency, hypothyroidism, toxic or drug-induced liver injury, long-term parenteral nutrition, intrahepatic biliary cysts, biliary atresia, abdominal mass, cholelithiasis, and pancreaticobiliary anomalies [15,16,17]. The following tests may be required: abdominal ultrasound with an evaluation of the bile ducts, hepatobiliary scintigraphy, liver biopsy, intraoperative cholangiogram and measurement of cortisol, thyroid stimulating hormone (TSH), thyroid hormones, electrolytes, acid-base parameters, and activity of galactose-1-phosphate uridyl transferase (GALT) in red blood cells. Other tests that should be considered include serum profile of transferrin isoforms, urinary bile acid profile, lipid panel, quantitative determination of amino acids in serum, urine reducing substance test, and urine tests based on GC–MS [15,16].

The results of performed tests in the described case demonstrated hemolytic disease as the sole cause of cholestasis. Galactosemia, tyrosinemia, and alpha-1 antitrypsin deficiency were excluded. TANDEM MS, GC–MS, and cystic fibrosis screening tests showed no abnormalities. Cortisol levels were within normal range. Congenital infection and viral infection were excluded. Abdominal ultrasound was used to rule out extrahepatic causes of cholestasis.

Cholestasis is more frequent in neonates with HDFN, especially in patients requiring intrauterine transfusions due to anemia. Low birth weight, neonatal anemia, elevated total serum bilirubin after birth, and elevated serum ferritin are also associated with an increased risk of cholestasis [7,11]. In the described case, all the above-mentioned risk factors were present.

Treatment of hyperferritinemia related to HDFN is not always necessary, considering that the need for red blood cell transfusion is transient [18]. According to the current literature, bilirubin levels in children with hemolytic disease usually normalize spontaneously within 1 week to 3 months. Ferritin levels also gradually decrease. Monitoring patients is recommended until biochemical parameters stabilize [7]. However, this may not always be the case as iron chelation therapy may be required in patients with severe complications of iron overload [7]. Often, the decision to use such treatment is supported by a liver biopsy, which is usually repeated after therapy [5,19,20,21]. Chelation therapy is the preferred medical treatment for anemia requiring multiple transfusions, e.g., Diamond-Blackfan anemia, sickle cell anemia, thalassemia, etc. [8,22]. In Table 2, we summarized all the described cases of using chelation therapy (deferoxamine) in children with HFDN [5,7,19,20,21,23].

Deferoxamine binds free Fe^3+^ cations, resulting in the formation of water-soluble ferrioxamine. This bound form is then excreted by kidneys, causing the characteristic red color of the urine and bile present in the feces. The lowest effective dose is generally used. Serum iron levels and response should be monitored to tailor dosage. Monitoring ferritin levels and renal and hepatic function is recommended during treatment [8].

It should be noted that liver dysfunction in children with HDFN may also be caused by hypoxia—due to anemia—and increased liver hematopoiesis [7].

In addition to the curative treatment of cholestasis, it is also essential to improve the nutritional status. Partial replacement of long-chain triglycerides (LCT) with medium-chain triglycerides (MCT) should be considered [24]. Increasing daily calorie supply is crucial due to malabsorption. Symptomatic treatment with ursodeoxycholic acid and fat-soluble vitamins (A, D, E, K) is also indicated [15,24].

Our patient was treated with ursodeoxycholic acid and fat-soluble vitamins. In the first year of life, the hypercaloric formula was also used to achieve appropriate body weight.

Despite severe complications of HDFN, especially liver failure with coagulation disorders, hypoalbuminemia, and cholestasis, the patient is developing normally, and the laboratory results do not show any significant abnormalities. This brings us to the conclusion that iron chelation therapy in children with hemolytic disease can improve liver function and prevent serious complications.

## 5. Conclusions

This case report shows that HDFN may cause severe symptomatic liver dysfunction due to iron overload requiring chelation therapy. In patients with life-threatening complications, appropriate medical care may result in normal development with no long-term consequences. In light of this case report, chelating agents may be considered as a treatment option in HDFN-associated liver failure. However, further research is necessary.

It is noteworthy that this study has some limitations. First, the literature review covered only the PubMed database, while a systematic search in the Scopus, MEDLINE, or ScienceDirect may have brought additional information on the subject. In addition, some data from medical history might be missing due to multicenter management.

## Figures and Tables

**Figure 1 jcm-12-07645-f001:**
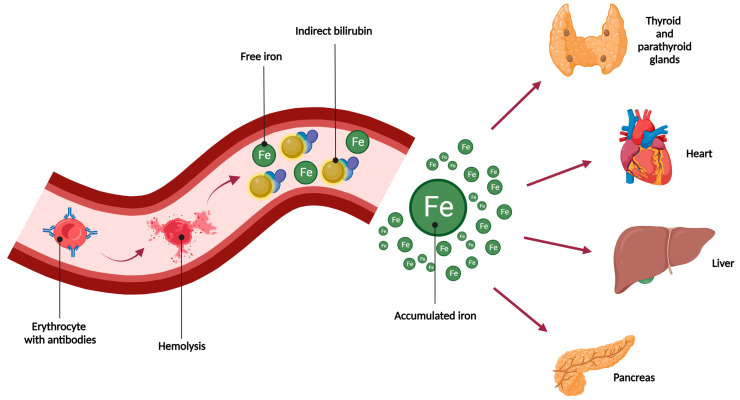
Causes and effects of iron accumulation [8].

**Table 1 jcm-12-07645-t001:** Extreme laboratory results in the first 3 months of life.

**Highest results of coagulation essays**	**Parameter**	**Value**	**Day of Life**
	aPTT	157.5 s	2
	INR	4.87	2
	PT	48.5 s	2
	thrombin time	55.3 s	2
	D-dimer	>128,000 ng/mL FEU *- with interpretative comment: estimated value 240,000 ng/mL FEU	3
**Lowest results of blood count**	ANC	648	26
	Hgb	6.1 g/dL	52
	Hct	18%	52
	PLT	14 × 10^9^/L	4
	reticulocyte count	0.12%	26
	absolute reticulocyte count	0.005 × 10^6^/µL	26
**Highest results of biochemical tests**	total bilirubin	43.55 mg/dL	54
	direct bilirubin	33.14 mg/dL	47
	bile acids	118.4 µmol/L	85
	AFP	172,405.4 IU/mL	71
	AST	3094 U/L	3
	ALT	373 U/L	3
	GGT	80 U/L	78
	ferritin	>33,511.2 ng/mL *	6
	creatinine	2.45 mg/dL	4

* Values above the detection range of the test. aPTT—activated partial thromboplastin time; INR—international normalized ratio; PT—prothrombin time; ANC—absolute neutrophil count; Hgb—hemoglobin; Hct—hematocrit; PLT—platelet count; AFP—alpha-fetoprotein; AST—aspartate aminotransferase; ALT—alanine aminotransferase; GGT—gamma-glutamyl transferase.

**Table 2 jcm-12-07645-t002:** List of described cases of use of chelation therapy in patients with HDFN.

First Author	Year of Publication	GA	BGS	IUT	ET	TUT	F [ng/mL]	DB [mmol/L]
Sreenan [21]	2000	33	Rh	5	1	1	4031	335
Yilmaz [20]	2006	34	Rh	2	2	4	5527	274
Demircioğlu [5]	2010	33	Rh	2	0	4	8842	212
Yalaz [23]	2011	34	Rh	2	1	3	28,800	479
Smits-Wintjens [7]	2012	n.d.	Rh	n.d.	n.d.	6	73,000	600
Khdair Ahmad [19]	2018	36	Rh	4	0	2	40,000	145
Drozdowska-Szymczak	2023	33	Rh	6	0	7	>33,511.2 ng/mL *	567

* Value above the detection range of the test. n.d.—no data; GA—gestational age; BGS—blood group system; IUT—number of intrauterine transfusions; ET—number of exchange transfusions, TUT—number of top-up transfusions; F—the highest ferritin level observed; DB—the highest direct bilirubin level observed.

## Data Availability

The data presented in this study are available in Table 1.

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
