# Peer review of "Liver Dysfunction with Severe Cholestasis and Coagulation Disorders in the Course of Hemolytic Disease of the Newborn Requiring Chelation Therapy—A Case Report and Review of the Literature"

_jcm, 2023, doi:10.3390/jcm12247645_

Round 1

Reviewer 1 Report

Comments and Suggestions for Authors

Dear Authors, 

thank you for the possibility of reviewing this manuscript. The topic is very interesting and could be of interest to readers.

I think the Introduction could be improved as it is pretty short. 

I do believe the manuscript needs some additional work. As this is a case report I believe you don't need the section Materials and Methods.

The Results and Discussion section is well-written and does not need revisions.

However, the Conclusions section could be improved as it consists of just one sentence.

Comments on the Quality of English Language

The quality of English of the manuscript is well and some minor changes are required. 

Reviewer 2 Report

Comments and Suggestions for Authors

The article is very important and interesting however I would like to comment to add graphical abstract.

Limitations and yours future perspectives may be added.

Grammar is weak needs improvement 

Comments on the Quality of English Language

Needs rechecking before publishing 

Reviewer 3 Report

Comments and Suggestions for Authors

Unfortunately, we also had some cases of HDFN and the outcome was variable, from intrauterine death to perinatal death to discharge followed by a satisfying evolution. Intrauterine transfusion will help improve the outcome but the risk of heart failure increases with the number of intruterine transfusions.

This paper covers most of the subject of HDFN and it is well-written

I suggest adding some info about the genetic implications of this disease. Also, genetic counselling should be addressed for these cases.

The conclusion paragraph is too short and needs to bemore detailed about your contributions to the field.

Comments on the Quality of English Language

English is fine

Reviewer 4 Report

Comments and Suggestions for Authors

Thank you for sending me this work so that I can evaluate it.

However, there must be some corrections that need to be made.

1-      Extra hepatic cholestasis was excluded in this patient. Reason? What radiological study was conducted to investigate extrahepatic cholestasis?

2-      Didn't serum calcium decrease in a patient who had so many transfusions?

3-      How did they come up with a solution for the reduced serum calcium?

4-      How many parenteral nutrition applications did you apply during your stay in the intensive care unit? Please specify it in the manuscript.

5-      Which lipids were used in parenteral nutrition? Were the lipids used related to cholestasis?

Round 2

Reviewer 1 Report

Comments and Suggestions for Authors

I would like to thank the authors for taking in mind my suggestions and reviewing the manuscript. 

I do believe the manuscript is siutable for publication.

Reviewer 3 Report

Comments and Suggestions for Authors

Thank you for responding to my suggestions. I see that the paper has been updated according to my suggestions.

The paper meets all the criteria to be published.

The conclusions are consistent with the evidence and arguments presented and they address the main question posed. The references are appropriate. Congratulations on the result!